# Analysis of Self-Gravitating Fluid Instabilities from the Post-Newtonian Boltzmann Equation

**DOI:** 10.3390/e26030246

**Published:** 2024-03-10

**Authors:** Gilberto M. Kremer

**Affiliations:** Departamento de Física, Universidade Federal do Paraná, Curitiba 81531-980, Brazil; kremer@fisica.ufpr.br

**Keywords:** self-gravitating fluid instability, Boltzmann equation, post-Newtonian theory

## Abstract

Self-gravitating fluid instabilities are analysed within the framework of a post-Newtonian Boltzmann equation coupled with the Poisson equations for the gravitational potentials of the post-Newtonian theory. The Poisson equations are determined from the knowledge of the energy–momentum tensor calculated from a post-Newtonian Maxwell–Jüttner distribution function. The one-particle distribution function and the gravitational potentials are perturbed from their background states, and the perturbations are represented by plane waves characterised by a wave number vector and time-dependent small amplitudes. The time-dependent amplitude of the one-particle distribution function is supposed to be a linear combination of the summational invariants of the post-Newtonian kinetic theory. From the coupled system of differential equations for the time-dependent amplitudes of the one-particle distribution function and gravitational potentials, an evolution equation for the mass density contrast is obtained. It is shown that for perturbation wavelengths smaller than the Jeans wavelength, the mass density contrast propagates as harmonic waves in time. For perturbation wavelengths greater than the Jeans wavelength, the mass density contrast grows in time, and the instability growth in the post-Newtonian theory is more accentuated than the one of the Newtonian theory.

## 1. Introduction

The search for self-gravitating fluid instabilities is an interesting subject for the determination of the structure formation of interstellar gas clouds. The first analysis of the self-gravitating fluid instabilities from the hydrodynamic equations coupled with the Newtonian Poisson equation was performed by Jeans [1] in 1902. In this work, he obtained a wavelength cutoff (Jeans wavelength) where perturbations with wavelengths smaller than the Jeans wavelength propagate as harmonic waves in time, while perturbations with wavelengths larger than the Jeans wavelength grow or decay in time. The gravitational collapse of self-gravitating interstellar gas clouds, which is connected with the exponential growth in time of the mass density perturbations, is known as the Jeans instability [2,3,4]. From a physical point of view, we may say that a collapse of a mass density inhomogeneity happens whenever the outwards pressure force of the self-gravitating gas cloud is smaller than the inwards gravitational force.

There are two main methods to treat the problem of self-gravitating fluid instabilities, one by using the hydrodynamic equations coupled with the Newtonian Poisson equation and another to consider the collisionless Boltzmann equation coupled with the Newtonian Poisson equation (see, for example, [3,4]).

The self-gravitating fluid instabilities were also considered in the f(R) gravity theory [5,6,7], where a modified dispersion relation implies new unstable modes.

Recently, the self-gravitating fluid instabilities were examined within the framework of the hydrodynamic equations and Poisson equations, which follow from the post-Newtonian theory (see [8,9]) in [10,11,12]. The same problem was analysed within the framework of a post-Newtonian collisonless Boltzmann equation (see [13,14,15]) in [16,17]. In these works, it was shown that the mass which is necessary for an overdensity to begin the gravitational collapse in the post-Newtonian theory is smaller than the one given by the Newtonian theory.

In the present work, we reexamine the self-gravitating fluid instabilities from the post-Newtonian collisionless Boltzmann equation coupled with the post-Newtonian Poisson equations in order to determine the difference between the Newtonian and post-Newtonian theories in the growth of the instabilities dictated by the mass density contrast. It is shown that the growth of the mass density contrast in the post-Newtonian theory is more accentuated than that of the Newtonian theory. The methodology used here differs from the one given in [16] since the amplitudes of the gravitational potentials and one-particle distribution function perturbations are considered to be time-dependent. Furthermore, the time-dependent amplitude of the one-particle distribution function is considered a linear combination of the summational invariants from the post-Newtonian kinetic theory.

The structure of the paper is the following: in Section 2, we introduce the post-Newtonian expressions for the Boltzmann and Poisson equations and the energy–momentum tensor defined in terms of the one-particle distribution function. The perturbations of the one-particle distribution function and gravitational potentials from a stationary equilibrium background is the subject of Section 3. In Section 4, the perturbations are represented by plane waves with small time-dependent amplitudes, and from the system of differential equations for the amplitudes, a time-evolution differential equation for the mass density contrast is found. The analysis of the solutions of the time-evolution differential equation for the mass density contrast is the topic of Section 5. The summary and conclusions of the work are given in Section 6 and Appendix A close the work.

## 2. Post-Newtonian Boltzmann Equation

The statistical description of a self-gravitating gas can be described by the Boltzmann equation, which governs the space-time evolution of the one-particle distribution function f(x,v,t) defined in the phase space spanned by the spatial coordinates x and velocity v of the particles. Here, we are interested in analysing the collisionless post-Newtonian Boltzmann equation (see [9] and the reference therein), which can be written in the first post-Newtonian approximation as [16]
(1)$$\begin{array}{c} \frac{\partial f}{\partial t}+v_{i}\frac{\partial f}{\partial x^{i}}+\frac{\partial U}{\partial x^{i}}\frac{\partial f}{\partial v^{i}}+\frac{1}{c^{2}}[\left(v^{2}-4U\right)\frac{\partial U}{\partial x^{i}}-4v_{i}v_{j}\frac{\partial U}{\partial x^{j}}\\ -3v_{i}\frac{\partial U}{\partial t}+2\frac{\partial \Phi }{\partial x^{i}}+\frac{\partial \Pi_{i}}{\partial t}+v_{j}\left(\frac{\partial \Pi_{i}}{\partial x^{j}}-\frac{\partial \Pi_{j}}{\partial x^{i}}\right)]\frac{\partial f}{\partial v^{i}}=0. \end{array}$$

Above $v_{i}$ is the three-velocity component of the particle four-velocity $u^{\mu }$ in first post-Newtonian approximation [8]
(2)$$\begin{array}{c} u^{0}=c\left[1+\frac{1}{c^{2}}\left(\frac{v^{2}}{2}+U\right)+\frac{1}{c^{4}}\left(\frac{3v^{4}}{8}+\frac{5Uv^{2}}{2}+\frac{U^{2}}{2}+2\Phi -\Pi_{i}v_{i}\right)\right],\;u^{i}=\frac{u^{0}v^{i}}{c}. \end{array}$$

Furthermore, in (Equation 1), *U* and Φ are scalar gravitational potentials, while $\Pi_{i}$ is a vector gravitational potential. They are defined in the first post-Newtonian approximation of the metric tensor $g_{\mu \nu }$ components by [8]
(3)$$\begin{array}{c} g_{00}=1-\frac{2U}{c^{2}}+\frac{2}{c^{2}}\left(U^{2}-2\Phi \right),\;g_{0i}=\frac{\Pi_{i}}{c^{3}},\;g_{ij}=-\left(1+\frac{2U}{c^{2}}\right)\delta_{ij}. \end{array}$$

The gravitational potentials *U*, Φ and $\Pi_{i}$ satisfy Poisson equations, which are given in terms of the energy–momentum tensor split in orders of $1/c^{n}$ denoted by $\overset{\;\;\;\;\;n}{T^{\mu \nu }}$, namely
(4)$$\begin{array}{c} \nabla^{2}U=-\frac{4\pi G}{c^{2}}\;\overset{\;\;\;\;0}{T^{00}},\;\nabla^{2}\Phi =-2\pi G\left(\overset{\;\;\;\;\;2}{T^{00}}+\overset{\;\;\;\;2}{T^{ii}}\right), \end{array}$$
(5)$$\begin{array}{c} \nabla^{2}\Pi^{i}=-\frac{16\pi G}{c}\overset{\;\;\;\;1}{T^{0i}}+\frac{\partial^{2}U}{\partial t\partial x^{i}}. \end{array}$$

The energy–momentum tensor is defined in kinetic theory of gases by the one-particle distribution function [18]
(6)$$\begin{array}{c} T^{\mu \nu }=m^{4}c\int u^{\mu }u^{\nu }f\frac{\sqrt{-g}\;d^{3}u}{u_{0}}, \end{array}$$
where the first post-Newtonian approximation of the invariant integration element is given by [19]
(7)$$\begin{array}{c} \frac{\sqrt{-g}\;d^{3}u}{u_{0}}=\left\{1+\frac{1}{c^{2}}\left[2v^{2}+6U\right]\right\}\frac{d^{3}v}{c}. \end{array}$$

## 3. Perturbations of the One-Particle Distribution Function and Gravitational Potentials

We shall write the one-particle distribution function and the gravitational potentials as sums of background and perturbed terms, where the background terms correspond to an equilibrium state.

For a relativistic gas, the equilibrium state of the one-particle distribution function is given by the Maxwell–Jüttner distribution function [18]. For a stationary equilibrium background, where the hydrodynamic velocity vanishes the first post-Newtonian approximation of the Maxwell–Jüttner distribution function $f_{MJ}$ is [19]
(8)$$\begin{array}{c} f_{MJ}=f_{0}\left\{1-\frac{\sigma^{2}}{c^{2}}\left[\frac{15}{8}+\frac{3v^{4}}{8\sigma^{4}}+\frac{2Uv^{2}}{\sigma^{4}}\right]\right\},\;f_{0}=\frac{\rho_{0}}{m^{4}{\left(2\pi \sigma^{2}\right)}^{\frac{3}{2}}}e^{-\frac{v^{2}}{2\sigma^{2}}}. \end{array}$$

In the above equation, $f_{0}$ is the non-relativistic Maxwellian distribution function, which is a function of the particle rest mass *m*, mass density $\rho_{0}$, gas particle velocity v, and the dispersion velocity σ.

By denoting the background terms of the gravitational potentials by the subscript zero and the perturbed terms of the one-particle distribution function and gravitational potentials by the subscript 1, we write
(9)$$\begin{array}{c} \;\;\;f\left(x,v,t\right)=f_{MJ}\left(x,v,t\right)+f_{1}\left(x,v,t\right), \end{array}$$
(10)$$\begin{array}{c} U\left(x,t\right)=U_{0}\left(x\right)+U_{1}\left(x,t\right), \end{array}$$
(11)$$\begin{array}{c} \Phi \left(x,t\right)=\Phi_{0}\left(x\right)+\Phi_{1}\left(x,t\right), \end{array}$$
(12)$$\begin{array}{c} \Pi_{i}\left(x,t\right)=\Pi_{i}^{0}\left(x\right)+\Pi_{i}^{1}\left(x,t\right). \end{array}$$

From the substitution of the representations (Equation 9)–(Equation 12) into the Boltzmann Equation (Equation 1), we obtain an equation for the space-time evolution of the perturbed one-particle distribution function [16]
(13)$$\begin{array}{c} \frac{\partial f_{1}}{\partial t}+v_{i}\frac{\partial f_{1}}{\partial x^{i}}+\frac{\partial U_{1}}{\partial x^{i}}\frac{\partial f_{MJ}^{0}}{\partial v^{i}}-\frac{2v^{2}f_{0}}{\sigma^{2}c^{2}}\left(\frac{\partial U_{1}}{\partial t}+v_{i}\frac{\partial U_{1}}{\partial x^{i}}\right)+\frac{1}{c^{2}}[\left(v^{2}-4U_{0}\right)\frac{\partial U_{1}}{\partial x^{i}}\\ \;+2\frac{\partial \Phi_{1}}{\partial x^{i}}+\frac{\partial \Pi_{i}^{1}}{\partial t}-3v_{i}\frac{\partial U_{1}}{\partial t}-4v_{i}v_{j}\frac{\partial U_{1}}{\partial x^{j}}+v_{j}\left(\frac{\partial \Pi_{i}^{1}}{\partial x^{j}}-\frac{\partial \Pi_{j}^{1}}{\partial x^{i}}\right)]\frac{\partial f_{0}}{\partial v^{i}}=0. \end{array}$$

Above, the background Maxwell–Jüttner distribution function is denoted by $f_{MJ}^{0}$. Here, we note that the background equation derived from (Equation 1) is identically satisfied when $\nabla U_{0}=0$, $\nabla \Phi_{0}=0$ and $\nabla \Pi_{i}^{0}=0$.

The conditions of vanishing background potential gravitational gradients do not satisfy the Poisson equations, and as usual, we have to assume “Jeans swindle” (see [4]), where the Poisson Equations (Equation 4) and (Equation 5) are valid only for the perturbed distribution function and gravitational potentials. Hence, from (Equation 4) and (Equation 5) together with the (Equation 6) and the representations (Equation 9)–(Equation 12) yield
(14)$$\begin{array}{c} \nabla^{2}U_{1}=-4\pi Gm^{4}\int f_{1}d^{3}v, \end{array}$$
(15)$$\begin{array}{c} \;\;\;\;\nabla^{2}\Pi_{1}^{i}=-16\pi Gm^{4}\int v_{i}f_{1}d^{3}v+\frac{\partial^{2}U_{1}}{\partial t\partial x^{i}}, \end{array}$$
(16)$$\begin{array}{c} \nabla^{2}\Phi_{1}=-2\pi Gm^{4}\int \left[\left(4v^{2}+8U_{0}\right)f_{1}-\left(\frac{2v^{2}}{\sigma^{2}}-8\right)U_{1}f_{0}\right]d^{3}v. \end{array}$$

## 4. Representation of Perturbations as Plane Waves

To go further in the analysis of the instabilities, we represent the perturbations as plane waves of wave number vector k and time-dependent small amplitudes ${\bar{f}}_{1}\left(v,t\right)$, ${\bar{U}}_{1}\left(t\right)$, ${\bar{\Phi }}_{1}\left(t\right)$ and $\bar{\Pi_{1}^{i}}\left(t\right)$, namely
(17)$$\begin{array}{c} f_{1}\left(x,v,t\right)={\bar{f}}_{1}\left(v,t\right)\;e^{i(k\cdot x)},\;U_{1}\left(x,t\right)={\bar{U}}_{1}\left(t\right)\;e^{i(k\cdot x)}, \end{array}$$
(18)$$\begin{array}{c} \;\Phi_{1}\left(x,t\right)={\bar{\Phi }}_{1}\left(t\right)\;e^{i(k\cdot x)},\;\Pi_{1}^{i}\left(x,t\right)=\bar{\Pi_{i}^{1}}\left(t\right)\;e^{i(k\cdot x)}. \end{array}$$

From the insertion of the plane wave representations (Equation 17) and (Equation 18) into the perturbed Boltzmann Equation (Equation 13), we obtain
(19)$$\begin{array}{c} f_{0}\left(iv\cdot k-\frac{d}{dt}\right){\bar{f}}_{1}-\frac{f_{0}}{\sigma^{2}}\{i\left(v\cdot k\right){\bar{U}}_{1}\left[1-\frac{\sigma^{2}}{c^{2}}\left(\frac{15}{8}+\frac{3v^{4}}{8\sigma^{4}}-\frac{v^{2}}{2\sigma^{2}}+\frac{2v^{2}U_{0}}{\sigma^{4}}\right)\right]\\ -\frac{1}{c^{2}}\left[v^{2}\frac{d{\bar{U}}_{1}}{dt}-2i\left(v\cdot k\right){\bar{\Phi }}_{1}-v_{i}\frac{d\bar{\Pi_{i}^{1}}}{dt}\right]\}=0, \end{array}$$
while the Poisson Equations (Equation 14)–(Equation 16) with the plane wave representations (Equation 17) and (Equation 18) become
(20)$$\begin{array}{c} \kappa^{2}{\bar{U}}_{1}=4\pi Gm^{4}\int {\bar{f}}_{1}d^{3}v, \end{array}$$
(21)$$\begin{array}{c} \;\;\;\;\;\kappa^{2}\bar{\Pi_{i}^{1}}=16\pi Gm^{4}\int v_{i}{\bar{f}}_{1}d^{3}v-ik_{i}\frac{d{\bar{U}}_{1}}{dt}, \end{array}$$
(22)$$\begin{array}{c} \kappa^{2}{\bar{\Phi }}_{1}=8\pi Gm^{4}\int \left(v^{2}+2U_{0}\right){\bar{f}}_{1}d^{3}v+4\pi G\rho_{0}{\bar{U}}_{1}. \end{array}$$

Now, we follow the methodology developed in [9] and consider the perturbed one-particle distribution function ${\bar{f}}_{1}\left(t\right)$ proportional to a sum of summational invariants, which are quantities that are conserved in the binary collision of the particles. In the relativistic kinetic theory of gases, the summational invariants are the rest mass of a particle *m* and the momentum four-vector $p^{\mu }$. Here, we write the perturbed one-particle distribution function as a linear combination of the summational invariants $\bar{A}\left(t\right)+{\bar{B}}_{\mu }\left(t\right)p^{\mu }$, where $\bar{A}\left(t\right)$ and ${\bar{B}}_{\mu }\left(t\right)$ are unknowns that do not depend on the momentum four-vector $p^{\mu }$.

The post-Newtonian approximation of the linear combination of the summational invariants $\bar{A}\left(t\right)+{\bar{B}}_{\mu }\left(t\right)p^{\mu }$ obtained from (Equation 2) and (Equation 3) and by considering terms up to the order $1/c^{4}$ reads
(23)$$\begin{array}{c} \bar{A}\left(t\right)+g_{\mu \nu }{\bar{B}}^{\mu }\left(t\right)p^{\mu }=\bar{A}\left(t\right)+m{\bar{B}}^{i}\left(t\right)\left\{\frac{\Pi_{i}^{0}}{c^{2}}-v_{i}\left[1+\frac{1}{c^{2}}\left(\frac{v^{2}}{2}+3U_{0}\right)\right]\right\}\\ +mc\left[1+\frac{1}{c^{2}}\left(\frac{v^{2}}{2}-U_{0}\right)+\frac{1}{c^{4}}\left(\frac{3v^{4}}{8}+\frac{3v^{2}U_{0}}{2}+\frac{U_{0}^{2}}{2}-6\Phi_{0}\right)\right]{\bar{B}}^{0}\left(t\right)\\ =A\left(t\right)+v^{2}\left[1+\frac{1}{c^{2}}\left(\frac{3v^{2}}{4}+3U_{0}\right)\right]D\left(t\right)+v_{i}\left[1+\frac{1}{c^{2}}\left(\frac{v^{2}}{2}+3U_{0}\right)\right]B_{i}\left(t\right), \end{array}$$
where new unknowns A(t), D(t) and $B_{i}\left(t\right)$ were introduced defined by
(24)$$\begin{array}{c} A\left(t\right)=\bar{A}\left(t\right)+mc{\bar{B}}^{0}\left(t\right)\left[1-\frac{U_{0}}{c^{2}}+\frac{1}{c^{4}}\left(\frac{U_{0}^{2}}{2}-6\Phi_{0}\right)\right]+\frac{m}{c^{2}}\Pi_{i}^{0}{\bar{B}}^{i}\left(t\right), \end{array}$$
(25)$$\begin{array}{c} \;\;\;\;\;\;\;\;\;\;D\left(t\right)=\frac{m{\bar{B}}^{0}\left(t\right)}{2c},\;B_{i}\left(t\right)=-m{\bar{B}}^{i}\left(t\right). \end{array}$$

From (Equation 23), we can identify the summational invariants in the post-Newtonian approximation, namely
(26)$$\begin{array}{c} 1,\;v^{2}\left[1+\frac{1}{c^{2}}\left(\frac{3v^{2}}{4}+3U_{0}\right)\right],\;v_{i}\left[1+\frac{1}{c^{2}}\left(\frac{v^{2}}{2}+3U_{0}\right)\right]. \end{array}$$

By neglecting the terms of order $1/c^{2}$, the above summational invariants reduce to those of the non-relativistic theory: 1, $v^{2}$ and $v_{i}$.

The representation of the perturbed one-particle distribution function follows from the product of the background Maxwell–Jüttner distribution function $f_{MJ}^{0}$ and the linear combination of the summational invariants (Equation 23):(27)$$\begin{array}{c} {\bar{f}}_{1}\left(t\right)=f_{MJ}^{0}\left(\bar{A}\left(t\right)+{\bar{B}}_{\mu }\left(t\right)p^{\mu }\right)=f_{0}\{\left[1-\frac{\sigma^{2}}{c^{2}}\left(\frac{15}{8}+\frac{3v^{4}}{8\sigma^{4}}+\frac{2U_{0}v^{2}}{\sigma^{4}}\right)\right]A\left(t\right)\\ +v^{2}\left[1-\frac{\sigma^{2}}{c^{2}}\left(\frac{15}{8}+\frac{3v^{4}}{8\sigma^{4}}+\frac{2U_{0}v^{2}}{\sigma^{4}}-\frac{3v^{2}}{4\sigma^{2}}-\frac{3U_{0}}{\sigma^{2}}\right)\right]D\left(t\right)\\ +v_{i}\left[1-\frac{\sigma^{2}}{c^{2}}\left(\frac{15}{8}+\frac{3v^{4}}{8\sigma^{4}}+\frac{2U_{0}v^{2}}{\sigma^{4}}-\frac{v^{2}}{2\sigma^{2}}-\frac{3U_{0}}{\sigma^{2}}\right)\right]B_{i}\left(t\right)\}. \end{array}$$

The dependence of the perturbed gravitational potentials ${\bar{U}}_{1}$, ${\bar{\Phi }}_{1}$ and ${\bar{\Pi }}_{i}^{1}$ on the unknown amplitudes A(t), D(t) and $B_{i}\left(t\right)$ of the perturbed one-particle distribution function follows from the insertion of (Equation 27) into (Equation 20)–(Equation 22) and the integration of the resulting equations, yielding
(28)$$\begin{array}{c} {\bar{U}}_{1}=\frac{\kappa_{J}^{2}}{\kappa^{2}}\sigma^{2}\left\{\left[1-\frac{\sigma^{2}}{c^{2}}\left(\frac{15}{2}+\frac{6U_{0}}{\sigma^{2}}\right)\right]A\left(t\right)+3\sigma^{2}\left[1-\frac{\sigma^{2}}{c^{2}}\left(\frac{45}{4}+\frac{7U_{0}}{\sigma^{2}}\right)\right]D\left(t\right)\right\},\; \end{array}$$
(29)$${\bar{\Pi }}_{i}^{1}k_{i}=4\frac{\kappa_{J}^{2}}{\kappa^{2}}\sigma^{4}\left[1-\frac{\sigma^{2}}{c^{2}}\left(\frac{25}{2}+\frac{7U_{0}}{\sigma^{2}}\right)\right]B_{i}\left(t\right)k_{i}-i\frac{d{\bar{U}}_{1}\left(t\right)}{dt},$$
(30)$$\begin{array}{c} {\bar{\Phi }}_{1}=\frac{\kappa_{J}^{2}}{\kappa^{2}}\sigma^{2}\{{\bar{U}}_{1}\left(t\right)+6\sigma^{2}\left[1+\frac{2U_{0}}{3\sigma^{2}}-\frac{\sigma^{2}}{c^{2}}\left(15+\frac{15U_{0}}{\sigma^{2}}+\frac{4U_{0}^{2}}{\sigma^{4}}\right)\right]A\left(t\right)\\ \;+30\sigma^{4}\left[1+\frac{2U_{0}}{5\sigma^{2}}-\frac{\sigma^{2}}{c^{2}}\left(\frac{81}{4}+\frac{31U_{0}}{2\sigma^{2}}+\frac{14U_{0}^{2}}{5\sigma^{4}}\right)\right]D\left(t\right)\}. \end{array}$$

In the above equations, we introduced the modulus of the wave number vector $\kappa =\sqrt{k\cdot k}$ and the Jeans wavelength $\kappa_{J}=\sqrt{4\pi G\rho_{0}/\sigma }$. Note that (Equation 29) follows from the scalar product of the vector equation for ${\bar{\Pi }}_{i}^{1}$ by $k_{i}$.

Now, we introduce the perturbed distribution function (Equation 27) into the perturbed Boltzmann Equation (Equation 19) and multiply the resulting equation by each of the summational invariants given in (Equation 26). From the integration of the resulting equations, by taking into account the invariant element of integration (Equation 7), the following system of differential equations emerges:
(31)$$\begin{array}{c} \left[1-\frac{9\sigma^{2}}{2c^{2}}\right]\frac{dA(t)}{dt}+3\sigma^{2}\left[1-\frac{\sigma^{2}}{c^{2}}\left(\frac{25}{4}+\frac{U_{0}}{c^{2}}\right)\right]\frac{dD(t)}{dt}+\frac{3}{c^{2}}\frac{d{\bar{U}}_{1}\left(t\right)}{dt}\\ +i\sigma^{2}\left[1-\frac{\sigma^{2}}{c^{2}}\left(\frac{15}{2}+\frac{U_{0}}{c^{2}}\right)\right]B_{i}\left(t\right)k_{i}=0, \end{array}$$(32)$$\begin{array}{c} \left[1-\frac{\sigma^{2}}{c^{2}}\left(\frac{25}{4}+\frac{U_{0}}{c^{2}}\right)\right]\frac{dA(t)}{dt}+5\sigma^{2}\left[1-\frac{\sigma^{2}}{c^{2}}\left(8+\frac{2U_{0}}{c^{2}}\right)\right]\frac{dD(t)}{dt}+\frac{5}{c^{2}}\frac{d{\bar{U}}_{1}\left(t\right)}{dt}\\ +i\frac{5}{3}\sigma^{2}\left[1-\frac{\sigma^{2}}{c^{2}}\left(\frac{39}{4}+\frac{2U_{0}}{c^{2}}\right)\right]B_{i}\left(t\right)k_{i}=0, \end{array}$$(33)$$\begin{array}{c} \{\left[1-\frac{\sigma^{2}}{c^{2}}\left(\frac{15}{2}+\frac{U_{0}}{c^{2}}\right)\right]A\left(t\right)+5\sigma^{2}\left[1-\frac{\sigma^{2}}{c^{2}}\left(\frac{39}{4}+\frac{2U_{0}}{c^{2}}\right)\right]D\left(t\right)-\frac{2{\bar{\Phi }}_{1}\left(t\right)}{\sigma^{2}c^{2}}\\ -\frac{{\bar{U}}_{1}\left(t\right)}{\sigma^{2}}\left[1-\frac{\sigma^{2}}{c^{2}}\left(5+\frac{U_{0}}{\sigma^{2}}\right)\right]\}ik_{i}+\frac{dB_{i}\left(t\right)}{dt}\left[1-\frac{\sigma^{2}}{c^{2}}\left(5-2\frac{U_{0}}{\sigma^{2}}\right)\right]-\frac{1}{\sigma^{2}c^{2}}\frac{d{\bar{\Pi }}_{i}^{1}\left(t\right)}{dt}=0. \end{array}$$


If we combine (Equation 31) and (Equation 32), we obtain the following differential equation:(34)$$\begin{array}{c} \frac{d}{dt}\left(A\left(t\right)-\frac{15\sigma^{2}}{4c^{2}}D\left(t\right)\sigma^{2}\right)=0\;⟹\;A\left(t\right)=C+\frac{15\sigma^{2}}{4c^{2}}D\left(t\right)\sigma^{2}, \end{array}$$

Hence, we can express the unknown A(t) in terms of the unknown D(t) and without loss of generality we can choose the constant C=1.

Here, we introduce the mass density contrast δρ, which refers to a parameter that indicates where a local increase in the matter density happens. The density contrast is defined by the ratio of the perturbed and unperturbed mass densities, namely,
(35)$$\begin{array}{c} \delta \rho =\frac{\int m{\bar{f}}_{1}d^{3}v}{\rho_{0}}=\left[1-\frac{\sigma^{2}}{c^{2}}\left(\frac{15}{2}+\frac{6U_{0}}{\sigma^{2}}\right)\right]A\left(t\right)+3\sigma^{2}\left[1-\frac{\sigma^{2}}{c^{2}}\left(\frac{45}{4}+\frac{7U_{0}}{\sigma^{2}}\right)\right]D\left(t\right). \end{array}$$

An evolution equation for the mass density contrast can be obtained from the differentiation of (Equation 31) with respect to time by using the relationships (Equation 34) and (Equation 35) and the elimination of $dB_{i}\left(t\right)/dt$ from (Equation 33) and of the gravitational potentials ${\bar{U}}_{1}\left(t\right)$, ${\bar{\Pi }}_{i}^{1}\left(t\right)$ and ${\bar{\Pi }}_{i}^{1}\left(t\right)$ by considering (Equation 28), (Equation 29) and (Equation 30), respectively. After some arrangements and by introducing the dimensionless time $\tau =t\sqrt{4\pi G\rho_{0}}$, the wavelength of the perturbation λ=2π/κ and the Jeans wavelength $\lambda_{J}=2\pi /\kappa_{J}$, we obtain
(36)$$\begin{array}{c} \frac{d^{2}\delta \rho }{d\tau^{2}}+5\frac{\lambda_{J}^{2}}{\lambda^{2}}\left\{1-\frac{\sigma^{2}}{c^{2}}\left[9+\frac{4U_{0}}{\sigma^{2}}+\frac{\lambda^{2}}{5\lambda_{J}^{2}}\left(61+24\frac{U_{0}}{\sigma^{2}}\right)\right]\right\}\delta \rho \\ -4\frac{\lambda_{J}^{2}}{\lambda^{2}}\left\{1-\frac{\sigma^{2}}{c^{2}}\left[\frac{135}{8}+\frac{10U_{0}}{\sigma^{2}}+4\frac{\lambda^{2}}{\lambda_{J}^{2}}\left(3+\frac{U_{0}}{\sigma^{2}}\right)\right]\right\}=0. \end{array}$$

## 5. The Time Evolution of the Mass Density Contrast

The solution of the differential Equation (Equation 36) for the mass density contrast reads
(37)$$\begin{array}{c} \delta \rho =C_{1}\exp\left\{i\sqrt{5}\frac{\lambda_{J}}{\lambda }\sqrt{1-\frac{\sigma^{2}}{c^{2}}\left[9+\frac{4U_{0}}{\sigma^{2}}+\frac{\lambda^{2}}{5\lambda_{J}^{2}}\left(61+24\frac{U_{0}}{\sigma^{2}}\right)\right]}\tau \right\}\\ +C_{2}\exp\left\{-i\sqrt{5}\frac{\lambda_{J}}{\lambda }\sqrt{1-\frac{\sigma^{2}}{c^{2}}\left[9+\frac{4U_{0}}{\sigma^{2}}+\frac{\lambda^{2}}{5\lambda_{J}^{2}}\left(61+24\frac{U_{0}}{\sigma^{2}}\right)\right]}\tau \right\}\\ +\frac{4\left\{1-\frac{\sigma^{2}}{c^{2}}\left[\frac{135}{8}+\frac{10U_{0}}{\sigma^{2}}+4\frac{\lambda^{2}}{\lambda_{J}^{2}}\left(3+\frac{U_{0}}{\sigma^{2}}\right)\right]\right\}}{5\left\{1-\frac{\sigma^{2}}{c^{2}}\left[9+\frac{4U_{0}}{\sigma^{2}}+\frac{\lambda^{2}}{5\lambda_{J}^{2}}\left(61+24\frac{U_{0}}{\sigma^{2}}\right)\right]\right\}}, \end{array}$$
where $C_{1}$ and $C_{2}$ are arbitrary constants.

Let us analyse the numerical solutions of the mass density contrast differential Equation (Equation 36) for the cases where the wavelength of the perturbation is bigger than the Jeans wavelength $\lambda >\lambda_{J}$ and smaller $\lambda <\lambda_{J}$.

The mass density contrast differential Equation (Equation 36) was solved for the initial conditions δρ(0)=1 and dδρ/dτ(0)=0.1 (say), and the numerical solutions are plotted in Figure 1 and Figure 2 for different values of the ratio between the dispersion velocity and the light speed, namely, σ/c=0, which corresponds to the Newtonian theory and σ/c=0.02 and σ/c=0.04 to the post-Newtonian theory. Furthermore, in the evaluation of (Equation 36), it was supposed that $U_{0}\approx \sigma^{2}$, which can be justified by the virial theorem, where the Newtonian gravitational potential can be approximated with the square of the dispersion velocity.

In Figure 1, the time evolution of the mass density contrast δρ is shown for the case where the wavelength of the perturbation is bigger than the Jeans wavelength ($\lambda_{J}/\lambda =0.01$) and by considering different values of the ratio between the dispersion velocity and the light speed σ/c. We observe from this figure that in this case, the mass density contrast grows in dimensionless time τ. We may also infer from this figure that the growth of the instability in the post-Newtonian theory differs from the one of the Newtonian theory, the former growing more rapidly than the latter. Furthermore, the increase in the ratio σ/c implies a more accentuated growth of the mass density contrast with the dimensionless time. As was pointed out previously, the so-called Jeans instability is connected with the gravitational collapse of self-gravitating interstellar gas clouds, which are associated with the mass density perturbations, which grow exponentially in time [2,3,4].

The time evolution of the mass density is plotted in Figure 2 for the case where the perturbation wavelength is smaller than the Jeans wavelength ($\lambda_{J}/\lambda =5$), where different values of the ratio between the dispersion velocity and the light speed σ/c were considered. We conclude from this figure that the mass density contrast propagates as harmonic waves in time. Note that there is no remarkable difference between the Newtonian and post-Newtonian solutions for the values adopted here.

## 6. Conclusions

In this work, the self-gravitating fluid instabilities were analysed within the framework of the post-Newtonian Boltzmann equation. The post-Newtonian Boltzmann equation is coupled with three Poison equations for the post-Newtonian gravitational potentials, which are functions of the energy–momentum tensor and determined from the knowledge of the post-Newtonian Maxwell–Jüttner distribution function. The one-particle distribution function and the gravitational potentials were perturbed from their background states represented by plane waves characterised by a wave number vector and time-dependent small amplitudes. The time-dependent amplitude of the one-particle distribution function was supposed to be a linear combination of the summational invariants of the post-Newtonian kinetic theory. From the coupled system of differential equations for the time-dependent amplitudes of the one-particle distribution function and gravitational potentials, an evolution equation for the mass density contrast was obtained. It was shown that for perturbation wavelengths smaller than the Jeans wavelength, the mass density contrast propagates as harmonic waves in time. For perturbation wavelengths greater than the Jeans wavelength, the mass density contrast grows in time, and the instability growth in the post-Newtonian theory is more accentuated than that of the Newtonian theory.

This work improves the former works on Jeans instability, where it was shown that in the post-Newtonian theory, the mass necessary for an overdensity to begin the gravitational collapse is smaller than in the Newtonian theory. Here, we have shown that in the post-Newtonian theory, the instabilities grow more rapidly than those in the Newtonian theory.

## Figures and Tables

**Figure 1 entropy-26-00246-f001:**
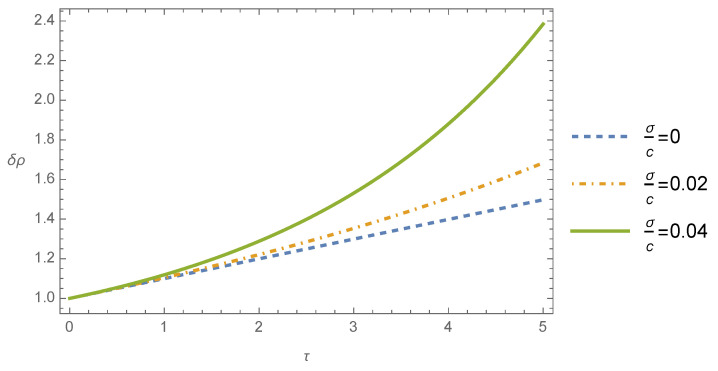
Mass density contrast δρ as a function of the dimensionless time τ for $\lambda_{J}/\lambda =0.01$ and different values of the ratio σ/c.

**Figure 2 entropy-26-00246-f002:**
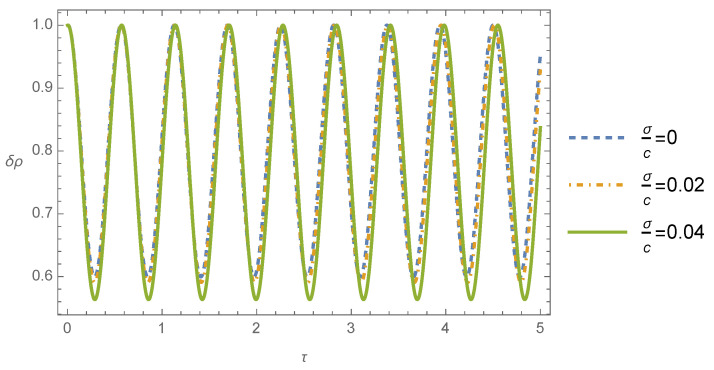
Mass density contrast δρ as a function of the dimensionless time τ for $\lambda_{J}/\lambda =5$ and different values of the ratio σ/c.

## Data Availability

No new data were created or analyzed in this study. Data sharing is not applicable to this article.

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
