# Peer review of "Analysis of Self-Gravitating Fluid Instabilities from the Post-Newtonian Boltzmann Equation"

_entropy, 2024, doi:10.3390/e26030246_

Round 1

Reviewer 1 Report

Comments and Suggestions for Authors

This paper deals with an interesting and so far poorly analyzed problem, namely the inclusion of post-Newtonian corrections in the Boltzmann equation. The presented analysis is robust and original and the results interesting. My only concern is on the presentation. Too often in reading the paper one needs to look at previous papers by the author to gain full understanding of the procedure and of the whole formalism. I would therefore ask the author to provide a more complete and self-consistent presentation, as to make the paper more comprehensible.

Author Response

Thank you for yours comments. 

I have introduced an appendix in order to fulfill your comments. The appendix is highlighted in red in the new version. 

Reviewer 2 Report

Comments and Suggestions for Authors

In the submitted manuscript, the author revisits the self-gravitating fluid instabilities derived from the post-Newtonian collisionless Boltzmann equation. The goal is to ascertain how the mass density fluctuations determine the growth of the instabilities in Newtonian and post-Newtonian theories: it is demonstrated that compared to the Newtonian theory, the post-Newtonian theory exhibits a more pronounced instability growth. The paper is well-written and clear, and the calculations are sound and explained in detail.  The paper can be accepted in its current form.

Author Response

Dear referee,

Thank you for your comments.